# Weakly supervised causal representation learning

**Johann Brehmer**[*]
Qualcomm AI Research[†]
jbrehmer@qti.qualcomm.com

**Pim de Haan**[*]
Qualcomm AI Research[†]
QUVA Lab, University of Amsterdam
pim@qti.qualcomm.com

**Phillip Lippe**
QUVA Lab, University of Amsterdam
p.lippe@uva.nl

**Taco Cohen**
Qualcomm AI Research[†]
tacos@qti.qualcomm.com

## Abstract

Learning high-level causal representations together with a causal model from unstructured low-level data such as pixels is impossible from observational data alone. We prove under mild assumptions that this representation is however identifiable in a weakly supervised setting. This involves a dataset with paired samples before and after random, unknown interventions, but no further labels. We then introduce implicit latent causal models, variational autoencoders that represent causal variables and causal structure without having to optimize an explicit discrete graph structure. On simple image data, including a novel dataset of simulated robotic manipulation, we demonstrate that such models can reliably identify the causal structure and disentangle causal variables.

## 1 Introduction

The dynamics of many systems can be described in terms of some high-level variables and causal relations between them. Often, these causal variables are not known but only observed in some unstructured, low-level representation, such as the pixels of a camera feed. Learning the causal representations together with the causal structure between them is a challenging problem and may be important for instance for applications in robotics and autonomous driving [1]. Without prior assumptions on the data-generating process or supervision, it is impossible to uniquely identify the causal variables and their causal structure [2, 3].

In this work, we show that a weak form of supervision is sufficient to identify both the causal representations and the structural causal model between them. We consider a setting in which we have access to data pairs, representing the system before and after a randomly chosen, unknown intervention while preserving the noise. This may approximate the generative process of data collected from a video feed of an external agent or demonstrator interacting with a system. Neither labels on the intervention targets nor active control of the interventions are

Figure 1: We learn to represent pixels $x$ as causal variables $z$. The bottom shows the effect of intervening on one variable. We prove that variables and causal model can be identified from samples $(x, \tilde{x})$.

---

[*]Equal contribution

[†]Qualcomm AI Research is an initiative of Qualcomm Technologies, Inc.

necessary for our identifiability theorem, making this setting useful for offline learning. We prove that with this form of weak supervision, and under certain assumptions (including that the interventions are stochastic and perfect and that all interventions occur in the dataset), latent causal models (LCMs)—structural causal models (SCMs) together with a decoder from the causal factors to the data space—are identifiable up to a relabelling and elementwise reparameterizations of the causal variables.

We then discuss two practical methods for LCM inference. First, we define explicit latent causal models (ELCMs) as a variational autoencoder (VAE) [4] in which the causal variables are the latent variables and the prior is based on an SCM. While this approach works in simple problems, it can be finicky and is difficult to scale. We trace this to a major challenge in causal representation learning, namely that it is a chicken-and-egg problem: it can be difficult to learn the causal variables when the causal graph is not yet learned, and it is difficult to learn the graph without knowing the variables.

To overcome this optimization difficulty, we introduce a second model class: implicit latent causal models (ILCMs). These models can represent causal structure and variables *without* requiring an explicit, discrete graph representation, which makes gradient-based optimization easier. Nevertheless, these models still contain the causal structure implicitly, and we discuss two algorithms that can extract it after the model is trained. Finally, we demonstrate ILCMs on synthetic datasets, including the new CausalCircuit dataset of a robot arm interacting with a causally connected system of light switches. We show that these models can robustly learn the true causal variables and the causal structure from pixels.

## 2   Related work

Our work builds on the work of Locatello et al. [5] on *disentangled representation learning*. The authors introduce a similar weakly supervised setting where observations are collected before and after unknown interventions. In contrast to our work, however, they focus on disentangled representations, i. e. (conditionally) independent factors of variation with a trivial causal graph, which our work subsumes as a special case. Other relevant works on disentangled representation learning and (nonlinear) independent component analysis include Refs. [6–12].

The problem of *causal representation learning* has been gaining attention lately, see the recent review by Schölkopf et al. [1]. Lu et al. [13] learn causal representations by observing similar causal models in different environments. von Kügelgen et al. [14] use the weakly supervised setting to study self-supervised learning, using a known but non-trivial causal graph between content and style factors. Lippe et al. [15] learn causal representations from time-series data from labelled interventions, assuming that causal effects are not instantaneous but can be temporally resolved. Yang et al. [16] propose to train a VAE with an SCM prior, but require the true causal variables as labels. Other relevant works include Refs. [17–21]. To the best of our knowledge, our work is the first to provide identifiability guarantees for arbitrary, unknown causal graphs in this weakly supervised setting.

## 3   Identifiability of latent causal models from weak supervision

In this section, we show theoretically that causal variables and causal mechanisms are identifiable from weak supervision. In Sec. 4 we will then demonstrate how we can learn causal models in practice by training a causally structured VAE.

### 3.1   Setup

We begin by defining latent causal models and the weakly supervised setting. Here, we only provide informal definitions and assume familiarity with common concepts from causality as introduced for instance in Ref. [22]. We provide a complete and precise treatment in Appendix A and discuss limitations of our setup and possible generalizations in Appendix B.

We describe the causal structure between latent variables as a Structural Causal Model (SCM). An SCM $\mathcal{C}$ describes the relation between causal variables $z_1, \ldots, z_n$ with domains $\mathcal{Z}_i$ and noise variables $\epsilon_1, \ldots, \epsilon_n$ with domains $\mathcal{E}_i$ along a directed acyclic graph (DAG) $\mathcal{G}(\mathcal{C})$. Causal mechanisms $f_i : \mathcal{E}_i \times \prod_{j \in \mathbf{pa}_i} \mathcal{Z}_j \to \mathcal{Z}_i$ describe how the value of a causal variable is determined from the associated noise variables, as well as the values of its parents in the graph. Finally, an SCM includes a probability measure for the noise variables.

An SCM entails a unique solution $s : \mathcal{E} \to \mathcal{Z}$ defined by successively applying the causal mechanisms. We require the causal mechanisms to be pointwise diffeomorphic, that is, for any value of the parents $z_{\mathbf{pa}_i}$ we have that $f_i(\cdot; z_{\mathbf{pa}_i})$ is invertible, differentiable, and its inverse is differentiable.[3] Then $s$ is also diffeomorphic and thus noise variables can be uniquely inferred from causal variables. This simplifies the weakly supervised distribution, as the only stochasticity comes from the noise variables and the intervention. The SCM also entails an observational distribution $p_{\mathcal{C}}(z)$ (Markov with respect to the graph of the SCM), which is the pushforward of $p_{\mathcal{E}}$ through the solution.

A perfect, stochastic intervention $(I, (\tilde{f}_i)_{i \in I})$ modifies an SCM by replacing for a subset of the causal variables, called the intervention target set $I \subset \{1, ..., n\}$, the causal mechanism $f_i$ with a new mechanism $\tilde{f}_i : \mathcal{E}_i \to \mathcal{Z}_i$, which does not depend on the parents. The intervened SCM has a new solution $\tilde{s}_I : \mathcal{E} \to \mathcal{Z}$. We call interventions atomic if the number of targeted variables is one or zero.

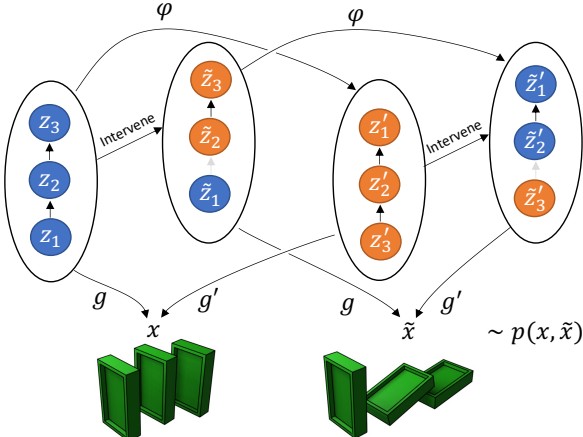

Figure 2: In LCM $\mathcal{M}$, $z_i$ denotes whether the $i$-th stone from the front is standing. Intervening on the second variable, $z_2$, leads to $\tilde{z}$. The decoder $g$ renders $z, \tilde{z}$ as images $x, \tilde{x}$. LCM $\mathcal{M}'$ has an equivalent representation in which $z_i'$ denotes whether the $i$-th stone from the *back* has *fallen*. In Thm. 1, we prove that if and only if two causal models have the same pixel distribution $p(x, \tilde{x})$, there exists an LCM isomorphism $\varphi$: an element-wise reparameterization of the causal variables plus a permutation of the ordering that commutes with interventions and causal mechanisms.

We will reason about generative models in a data space $\mathcal{X}$, in which the causal structure is latent. Also including a distribution of interventions (as in Ref. [23]), we define LCMs:

**Definition 1** (Latent causal model (LCM)). *A latent causal model $\mathcal{M} = \langle \mathcal{C}, \mathcal{X}, g, \mathcal{I}, p_{\mathcal{I}} \rangle$ consists of*

- *an acyclic SCM $\mathcal{C}$, which is faithful (all independencies are encoded in its graph [24]),*
- *an observation space $\mathcal{X}$,*
- *a decoder $g : \mathcal{Z} \to \mathcal{X}$ that is diffeomorphic onto its image,*
- *a set $\mathcal{I}$ of interventions on $\mathcal{C}$, and*
- *a probability measure $p_{\mathcal{I}}$ over $\mathcal{I}$.*

We define two LCMs as equivalent if all of their components are equal up to a permutation of the causal variables and elementwise diffeomorphic reparameterizations of each variable, see Fig. 2.

**Definition 2** (LCM isomorphism (informal)). *Let $\mathcal{M} = \langle \mathcal{C}, \mathcal{X}, g, \mathcal{I}, p_{\mathcal{I}} \rangle$ and $\mathcal{M}' = \langle \mathcal{C}', \mathcal{X}, g', \mathcal{I}', p_{\mathcal{I}'}' \rangle$ be two LCMs with identical observation space. An LCM isomorphism between them is a graph isomorphism $\psi : \mathcal{G}(\mathcal{C}) \to \mathcal{G}(\mathcal{C}')$ together with elementwise diffeomorphisms for noise and causal variables that tell us how to reparameterize them, such that the structure functions, noise distributions, decoder, intervention set, and intervention distribution of $\mathcal{M}'$ are compatible with the corresponding elements of $\mathcal{M}$ reparameterized through the graph isomorphism and elementwise diffeomorphisms. $\mathcal{M}$ and $\mathcal{M}'$ are equivalent, $\mathcal{M} \sim \mathcal{M}'$, if and only if there is an LCM isomorphism between them.*

Following Locatello et al. [5], we define a generative process of pre- and post-interventional data:[4]

**Definition 3** (Weakly supervised generative process). *Consider an LCM $\mathcal{M}$ where the underlying SCM has continuous noise spaces $\mathcal{E}_i$, independent probabilities $p_{\mathcal{E}_i}$, and admits a solution $s$. We*

---

[3]Under some mild smoothness assumptions, any SCM can be brought into this form by elementwise redefinitions of the variables, preserving the observational and interventional distributions, but not the weakly supervised/counterfactual distribution.

[4]This construction is closely related to twinned SCMs [25, Def. 2.17], typically used to compute counterfactual queries $p(\tilde{z}_{\setminus I} | z, \tilde{z}_I)$. We instead focus on the joint distribution of pre-intervention and post-intervention data.

*define the weakly supervised generative process of data pairs $(x, \tilde{x}) \sim p_{\mathcal{M}}^{\mathcal{X}}(x, \tilde{x})$ as follows:*

$$
\begin{aligned}
&\epsilon \sim p_{\mathcal{E}}\,, && && z = s(\epsilon)\,, && x = g(z)\,, \\
&I \sim p_{\mathcal{I}}\,, && \forall i \in I\,, \ \tilde{\epsilon}_i \sim p_{\tilde{\mathcal{E}}_i}\,, && \forall i \notin I\,, \ \tilde{\epsilon}_i = \epsilon_i\,, && \tilde{z} = \tilde{s}_I(\tilde{\epsilon})\,, && \tilde{x} = g(\tilde{z})\,. && (1)
\end{aligned}
$$

## 3.2 Identifiability result

The main theoretical result of this paper is that an LCM $\mathcal{M}$ can be identified from $p(x, \tilde{x})$ up to a relabeling and elementwise transformations of the causal variables:

**Theorem 1** (Identifiability of $\mathbb{R}$-valued LCMs from weak supervision). *Let $\mathcal{M} = \langle \mathcal{C}, \mathcal{X}, g, \mathcal{I}, p_{\mathcal{I}} \rangle$ and $\mathcal{M}' = \langle \mathcal{C}', \mathcal{X}, g', \mathcal{I}', p'_{\mathcal{I}'} \rangle$ be LCMs with the following properties:*

- *The LCMs have an identical observation space $\mathcal{X}$.*
- *The SCMs $\mathcal{C}$ and $\mathcal{C}'$ both consist of $n$ real-valued endogeneous causal variables and corresponding exogeneous noise variables, i. e. $\mathcal{E}_i = \mathcal{Z}_i = \mathcal{Z}'_i = \mathcal{E}'_i = \mathbb{R}$.*
- *The intervention sets $\mathcal{I}$ and $\mathcal{I}'$ consist of all atomic, perfect interventions, $\mathcal{I} = \{\emptyset, \{z_0\}, \ldots, \{z_n\}\}$ and similar for $\mathcal{I}'$.*
- *The intervention distribution $p_{\mathcal{I}}$ and $p'_{\mathcal{I}'}$ have full support.*

*Then the following two statements are equivalent:*

1. *The LCMs entail equal weakly supervised distributions, $p_{\mathcal{M}}^{\mathcal{X}}(x, \tilde{x}) = p_{\mathcal{M}'}^{\mathcal{X}}(x, \tilde{x})$.*
2. *The LCMs are equivalent in the sense of Def. 2, $\mathcal{M} \sim \mathcal{M}'$.*

Let us summarize the key steps of our proof, which we provide in its entirety in Sec. A in the supplementary material. The direction $2 \Rightarrow 1$ follows from the definition of equivalence. The direction $1 \Rightarrow 2$ is proven constructively along the following steps:

1. We begin by defining a diffeomorphism $\varphi = g'^{-1} \circ g : \mathcal{Z} \to \mathcal{Z}'$ and note that if $z, \tilde{z} \sim p_{\mathcal{C}}^{\mathcal{Z}}(z, \tilde{z})$, the weakly supervised distribution of causal variables of model $\mathcal{C}$, then $\varphi(z), \varphi(\tilde{z}) \sim p_{\mathcal{C}'}^{\mathcal{Z}'}(z', \tilde{z}')$. The distribution over $z, \tilde{z}$ is a mixture, where each intervention target $I$ gives a mixture component; each component is supported on a different $(n+1)$-dimensional submanifold. Therefore, there exists a bijection between the components $\psi : [n] \to [n]$ that maps intervention targets $I$ in $\mathcal{M}$ to intervention targets $I' = \psi(I)$ in $\mathcal{M}'$. Furthermore, because the joint distribution $z, \tilde{z}$ is preserved by $\varphi$, first mapping with $\varphi$, then intervening, $\mathcal{Z} \xrightarrow{\varphi} \mathcal{Z}' \xrightarrow{I'} \widetilde{\mathcal{Z}}'$, equals $\mathcal{Z} \xrightarrow{I} \widetilde{\mathcal{Z}} \xrightarrow{\varphi} \widetilde{\mathcal{Z}}'$.

2. Because $I = \{i\}$ is a perfect intervention, for the map $\mathcal{Z} \xrightarrow{\varphi} \mathcal{Z}' \xrightarrow{I'} \widetilde{\mathcal{Z}}'$, $\tilde{z}'_{i'}$ is independent of $z'$. Thus, in both maps, $\tilde{z}'_{i'}$ is independent of $z$. This means that for the path through $\widetilde{\mathcal{Z}}$, the intervention sample $\tilde{z}_i$ is transformed into $\tilde{z}'_{i'}$ independently of $z$. For $\mathbb{R}$-valued variables, this statistical independence implies that the transformation is constant in $z$, and thus $\varphi(z)_{i'}$ is constant in $z_j$ for $j \neq i$. $\varphi$ is therefore an elementwise reparametrization.

3. Using this, we can show that $\psi$ is a causal graph isomorphism and that it is compatible with the causal mechanisms. This proves LCM equivalence $\mathcal{M} \sim \mathcal{M}'$.

## 4 Practical latent causal models

Theorem 1 means that it is possible to learn causal structure from pixel-level data in the weakly supervised setting. Consider a system that is described by an unknown true LCM and assume that we have access to data pairs $(x, \tilde{x})$ sampled from its probability density. Then we can train another LCM with learnable components by maximum likelihood. Assuming sufficient data and perfect optimization, this model's density will match that of the ground-truth LCM. Our identifiability result guarantees that the trained LCM then has the same causal variables and causal structure as the ground truth, up to relabelling.

In the following, we describe two neural LCM implementations that can be trained on data.

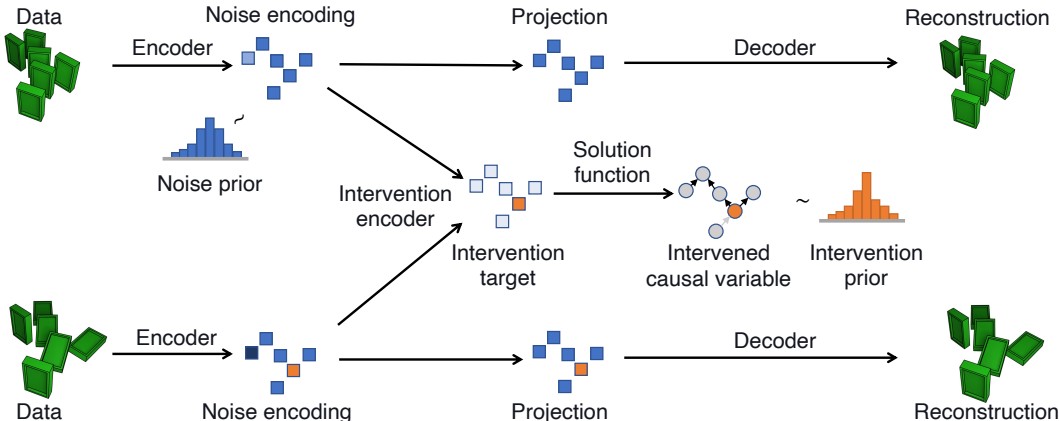

Figure 3: ILCM architecture. Pre- and post-intervention data (left) are encoded to noise encodings and intervention targets, which are then decoded back to the data space. To compute the prior probability density, the noise encodings are transformed into causal variables with the neural solution function.

## 4.1 Explicit latent causal models (ELCMs)

To implement LCMs with neural networks, we use the variational autoencoder (VAE) framework [4]. We first consider an approach where the causal variables $(z, \tilde{z})$ are the latent variables. Data $(x, \tilde{x})$ and latents $(z, \tilde{z})$ are linked by a stochastic encoder $q(z|x)$ and decoder $p(x|z)$; unlike the deterministic decoder from Sec. 3 this allows us to map high-dimensional data spaces (like images) to low-dimensional causal variables. The causal structure is encoded in the prior $p(z, \tilde{z})$ and consists of a learnable causal graph and learnable causal mechanisms $f_i$. The first contribution to the prior density is the observational probability density $p(z)$, which factorizes according to the causal graph into components $p(z_i|z_{\mathbf{pa}_i})$, which are given by fixed base densities and the causal mechanisms. The second contribution is the interventional conditional density $p(\tilde{z}|z)$, which is also computable from the graph and causal mechanisms. The model can be trained on the ELBO loss, a variational bound on $-\log p(x, \tilde{x})$. For more details, see Appendix E.

We call this an *explicit latent causal model* (ELCM), as the model directly parameterizes all components of an LCM. In particular, ELCMs contain an explicit representation of the causal graph and causal mechanisms. The graph can be learned by an exhaustive search over all DAGs or through a differentiable DAG parameterization [26–29] and gradient descent.

In our experiments with ELCMs, which we describe in Appendix E, we find that optimally trained ELCMs indeed correctly identify the causal structure and disentangle causal variables on simple datasets. However, jointly learning explicit graph and variable representations presents a challenging optimization problem. In particular, we observe that the loss landscape has local minima corresponding to wrong graph configurations.

## 4.2 Implicit latent causal models (ILCMs)

To enable causal representation learning in a more robust, scalable way, we propose a second LCM implementation: *Implicit Latent Causal Models* (ILCMs). Like ELCMs, ILCMs are also variational autoencoders with a causally structured prior. The key difference is that ILCMs represent the causal structure through neural solution functions $s(e)$. Under our assumption of diffeomorphic causal mechanisms, the solution function—which maps the vector of noise variables to the causal variables—contains the same information as the causal graph and causal mechanisms that we parameterize with neural networks in ELCMs (see Appendix C). However, unlike ELCMs, this parameterization does not require an explicit graph parameterization. In practice, ILCMs are thus easier to train than ELCMs.

**Latents** The latent variables in an ILCMs are *noise encodings*, defined through the inverse solution function as $e = s^{-1}(z)$ and $\tilde{e} = s^{-1}(\tilde{z})$. The pre-intervention noise encoding $e$ is identical to the SCM noise variables. The post-intervention noise encoding $\tilde{e}$ corresponds to the value of the SCM noise variables that would have generated the post-intervention causal variables $\tilde{z}$ under the

unintervened SCM mechanisms. ILCMs contain a stochastic encoder $q(e|x)$ and decoder $p(x|e)$ that map data $(x, \tilde{x})$ to noise encodings $(e, \tilde{e})$.

Noise encodings have the convenient property that under an intervention with intervention targets $I$, precisely the components $e_I$ change value: $e_i \neq \tilde{e}_i \Leftrightarrow i \in I$ with probability 1. We prove this property in Appendix A. This means that from noise encodings $e, \tilde{e}$, we can infer interventions easily. We use a simple heuristic intervention encoder that assigns higher intervention probability $q(i \in I|x, \tilde{x})$ to a component $i$ the more this component of the noise encoding changes under interventions:

$$\log q(i \in I|x, \tilde{x}) \sim h\big(\mu_e(x)_i - \mu_e(\tilde{x})_i\big), \tag{2}$$

where $\mu_e(x)$ is the mean function of the noise encoder $q(e|x)$ and $h$ is a quadratic function with learnable parameters. Both the equality pattern of $e$ under interventions and this heuristic intervention encoder are similar to the ones used for disentangled representation learning in Ref. [5].

**Prior**   Given encoders for noise encodings and intervention targets, let us now write down the prior $p(e, \tilde{e}, I)$, which encodes the structure of the weakly supervised setting. The intervention-target prior $p(I)$ and the pre-intervention noise distribution $p(e)$ are given by simple base densities, which we choose as uniform categorical and standard Gaussian, respectively. The post-intervention noise encodings $\tilde{e}$ follow the conditional probability distribution

$$p(\tilde{e}|e, I) = \prod_{i \notin I} \delta(\tilde{e}_i - e_i) \prod_{i \in I} p(\tilde{e}_i|e) = \prod_{i \notin I} \delta(\tilde{e}_i - e_i) \prod_{i \in I} \tilde{p}(\bar{z}_i) \left| \frac{\partial \bar{z}_i}{\partial \tilde{e}_i} \right|, \quad \bar{z}_i = \bar{s}_i(\tilde{e}_i; e_{\setminus i}). \tag{3}$$

In the second equality we have parameterized the conditional density $p(\tilde{e}_i|e)$ with a conditional normalizing flow consisting of a learnable diffeomorphic transformation $\tilde{e}_i \mapsto \bar{z}_i = \bar{s}_i(\tilde{e}_i; e_{\setminus i})$ and a base density $\tilde{p}$ on $\bar{z}_i$, which we choose as standard Gaussian.

How does this prior encode causal structure? We rely on three key properties of SCMs, shown in Appendix A: 1) the noise variables $e_i$ are independent of each other; 2) upon intervening on variable $i$, the post-intervention causal variable $\tilde{z}_i$ are independent of all $e_j$; 3) while for the other variables $j \neq i$, the noise encodings are unchanged $\tilde{e}_j = e_j$. These three properties are ensured in the ILCM prior in Eq. (3). We show in Appendix C that therefore each ILCM is equivalent to a unique ELCM. For each variable $i$, the ILCM function $\bar{s}_i$ is equal to the solution function $s_i$ of the equivalent ELCM, which maps from noise variables to causal variable $z_i$.

Thus, by learning to transform $\tilde{e}_i$ into $\tilde{z}_i = \bar{s}(\tilde{e}_i; e)$ in the ILCM, we learn the solution function of the corresponding ELCM. This implicitly describes both the causal graph and the causal mechanisms $f_i$. We can thus learn a causal model without ever explicitly modelling a graph.[5]

The final question is how to implement the first terms in Eq. (3), which encode that those noise encodings that are not part of the intervention targets $I$ should not change value under the intervention. We enforce this in the encoder by setting the non-intervention components of $e$ and $\tilde{e}$ to the same value [similar to 5]. In Appendix C this procedure is described in more detail. We will refer to this projective noise encoder as $q(e, \tilde{e}|x, \tilde{x}, I)$.

**Learning**   Putting everything together, an ILCM consists of an intervention encoder $q(I|x, \tilde{x})$, a noise encoder $q(e, \tilde{e}|x, \tilde{x}, I)$, a noise decoder $p(x|e)$, and transformations / solution functions $s_i(\cdot; e)$, see Fig. 3. All of these components are implemented with neural networks and learnable, see Appendix C for details. The lower bound on the joint log likelihood of pre-intervention and post-intervention data is given by

$$\log p(x, \tilde{x}) \geq \mathbb{E}_{I \sim q(I|x, \tilde{x})} \mathbb{E}_{e, \tilde{e} \sim q(e, \tilde{e}|x, \tilde{x}, I)} \bigg[ \log p(I) + \log p(e) + \log p(\tilde{e}|e, I)$$

$$- \log q(I|x, \tilde{x}) - \log q(e, \tilde{e}|x, \tilde{x}, I) + \log p(x|e) + \log p(\tilde{x}|\tilde{e}) \bigg]. \tag{4}$$

---

[5]The solution function of an ELCM only depends on ancestors in the graph. The learned transformation $s_i(e_i; e_{\setminus i})$ of an ILCM should thus also depend only on ancestors of $i$. As each $s_i$ is constructed to be a diffeomorphism in its first argument, jointly they have a triangular structure and thus a diffeomorphism $s : e \mapsto z$. In practice, however, the learned solution functions may still depend weakly on non-ancestors. Therefore, to ensure that $s$ always forms a diffeomorphism, at some point in training, we test functional dependence to infer ancestral dependence, pick a topological ordering of variables conforming to the ancestry, and parameterize the solution functions $s_i$ to only depend on earlier variables in the ordering.

The model is trained by minimizing the corresponding VAE loss, learning to map low-level data to noise variables (with $q$) and to map noise variables to causal variables (with $s$). The expectation over $I$ is computed via summation, but could alternatively be done with sampling.

In practice, we find it beneficial to add a regularization term to the loss that disincentivizes collapse to a lower-dimensional submanifold of the latent space. For each batch of training data, we compute the batch-aggregate intervention posterior $q_I(I) = \mathbb{E}_{x,\tilde{x}\in\text{batch}}[q(I|x,\tilde{x})]$. To the beta-VAE loss we then add the negative entropy of this distribution, weighted with a hyperparameter.

**Downstream tasks**   Despite the implicit representation of causal structure, we argue that ILCMs let us solve various tasks:

- *Causal representation learning / disentanglement*: ILCMs allow us to map low-level data $x$ to causal variables $z$ by applying the encoder $q$ followed by the solution functions $s$.
- *Intervention inference*: It is also straightforward to infer intervention targets from an observed pair $(x, \tilde{x})$ of pre-intervention and post-intervention data, as this just requires evaluating the intervention-target encoder $q_I(x, \tilde{x})$.
- *Causal discovery / identification*: We propose two methods to infer the causal graphs after training an ILCM. One is to use an off-the-shelf method for causal discovery on the learned representations. Since the ILCM allows us to infer intervention targets, we can use intervention-based algorithms. In this paper, we use ENCO [28], a recent differentiable causal discovery method that exploits interventions to obtain acyclic graphs without requiring constrained optimization. Alternatives to ENCO include DCDI [27] and GIES [30].
  Alternatively, we can analyze the causal structure implicitly represented in the learned solution functions $s_i$. We propose a heuristic algorithm that proceeds in three steps. First, it infers the topological order by sorting variables such that $s_i$ only depends on $e_j$ if $z_i$ is after $z_j$ in the topological order. It then iteratively rewrites the solution functions such that they only depend on ancestors in the topological order. Finally, it determines which causal ancestors are direct parents by testing the functional dependence of the causal mechanisms. We describe this algorithm in more detail in Appendix C.
- *Generation of interventions and counterfactuals*: The ILCM entails a generative model for pairs of pre- and post-intervention data. It is straightforward to sample from the joint distribution $p(x, \tilde{x}, I)$, from the conditional $p(x, \tilde{x}|I)$, or from the conditional $p(\tilde{x}|x, I)$.

## 5   Experiments

Finally, we demonstrate latent causal models in practice. Here we focus on implicit LCMs; explicit LCMs are demonstrated in similar experiments in Appendix E. We evaluate the causal graphs learned by the ILCM models either with ENCO (ILCM-E) or with the heuristic algorithm described above (ILCM-H).

**Baselines**   Since we are to the best of our knowledge the first to study causal representation learning in this weakly supervised setting, we are not aware of any baseline methods designed for this task. We nevertheless compare ILCMs to three other methods. First, we define a *disentanglement VAE* that models the weakly supervised process, but assumes independent factors of variation rather than a non-trivial causal structure between the variables. This baseline is similar to the method proposed by Ref. [5], but it differs in some implementation details to be more comparable to our ILCM setup. We infer the causal graph between the learned representations with ENCO (dVAE-E). We also compare to an unstructured $\beta$-VAE that treats $x$ and $\tilde{x}$ as i.i.d. and uses a standard Gaussian prior. Finally, for the pixel-level data, we consider a slot attention model [31], which segments the image unsupervisedly into as many objects as there are causal variables. The latent representation associated to each object is considered a learned causal variable.

### 5.1   2D toy experiment

We first demonstrate LCMs in a pedagogical toy experiment with $\mathcal{X} = \mathcal{Z} = \mathbb{R}^2$. Training data is generated from a nonlinear SCM with the graph $z_1 \rightarrow z_2$ and mapped to the data space through a randomly initialized normalizing flow.

An ILCM trained in the weakly supervised setting is able to reconstruct the causal factors accurately up

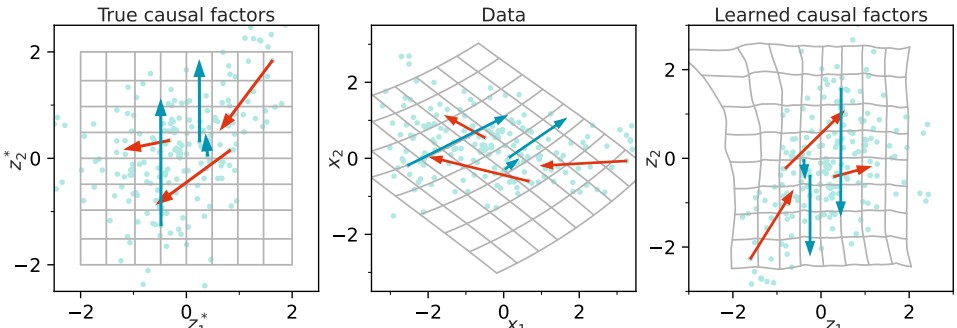

Figure 4: 2D toy data with graph $z_1^* \to z_2^*$. The grey grids show the map between true causal factors, data, and latent causal factors learned by the LCM. The mint dots indicate the observational data distribution, the arrows from $z$ to $\tilde{z}$ show interventions targeting $z_1^*$ (red) or $z_2^*$ (blue). The fact that axis-aligned lines in the true latent space are mapped to axis-aligned lines in the learned latent space implies that the disentanglement succeeded.

to elementwise reparameterizations, as shown in Fig. 4. In Tbl. 1 we quantify the quality of the learned representations with the DCI disentanglement score [32]. We find that our LCM is able to disentangle the causal factors almost perfectly, while the baselines, which assume independent factors of variation, fail as expected. Both the ILCM and the dVAE baseline infer the intervention targets with high accuracy. Finally, we test the quality of the learned causal graphs. We infer the implicit graph with ENCO and the heuristic algorithm discussed above. In both cases, the learned causal graph is identical to the correct one, whereas the representations found by the dVAE baseline induce a wrong graph.

## 5.2 Causal3DIdent

We then turn to pixel-level data and more complex causal graphs. We test ILCMs on an adaptation of the Causal3DIdent dataset [14], which contains images of three-dimensional objects under variable positions and lighting conditions. We consider three causal variables representing object hue, the spotlight hue, and the position of the spotlight. We construct six versions of this dataset, each with a different causal graph, randomly initialized nonlinear structure functions, and heteroskedastic noise. These are mapped to images with a resolution of $64 \times 64$, see Fig. 5 for examples.

ILCMs are again able to disentangle the causal variables reliably. The results in Tbl. 1 show that the learned representations are more disentangled than those learned by methods that do not account for causal structure. The LCM as well as the dVAE baseline can infer interventions with almost perfect accuracy. We demonstrate this in Fig. 5 by comparing true and inferred interventions, see Sec. D.3 of the supplementary material for details. The ILCMs also learn the causal graphs accurately, while the acausal dVAE-E baseline does in most cases not find the correct causal graphs.

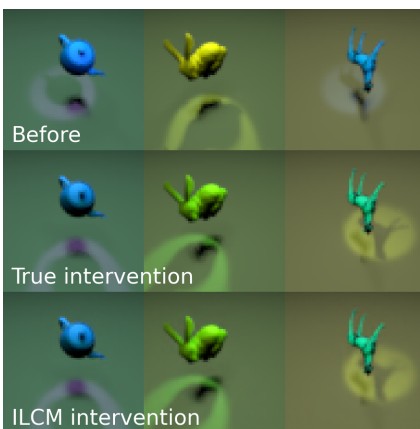

Figure 5: Causal3DIdent before (top) and after (middle) interventions, and post-intervention samples generated from the ILCM under the intervention inferred from the data (bottom), indicating we correctly learned to intervene.

## 5.3 CausalCircuit

While Causal3DIdent provides a good test of the ability to disentangle features that materialize in pixel space in different ways, like through the position of lights and the color of objects, the underlying causal structure we imposed may feel rather ad-hoc. To explore causal representation learning in a more intuitively causal setting, we introduce a new dataset, which we call CausalCircuit.

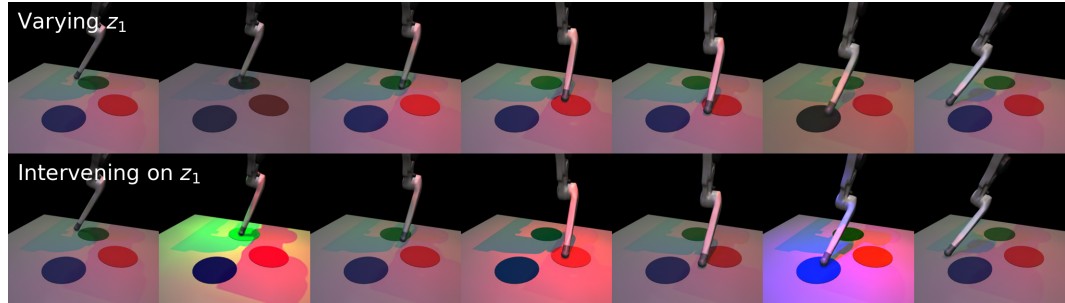

Figure 6: Varying learned causal factors vs. intervening on them. With a trained ILCM, we encode a single test image (left column). In the top row, we then vary the latent $z_1$ independently, without computing causal effects, and show the corresponding reconstructed images. Only the robot arm position changes, highlighting that we learned a disentangled representation. In the bottom row we instead *intervene* on $z_1$ and observe the causal effects: the robot arm may activate lights, which in turn can affect other lights in the circuit.

The CausalCircuit system consists of a robot arm that can interact with multiple touch-sensitive lights. The lights are connected with a stochastic circuit: a light is more likely to be on if its button is pressed or if its parent lights are on. The robot arm itself can be seen as part of the causal system. Concretely, we consider the causal graph shown in Fig. 7. This system is observed from a fixed-position camera, and we generate samples in $512 \times 512 \times 3$ resolution with MuJoCo [33], see Sec. D.4 of the supplementary material for more details.

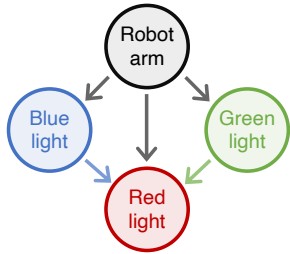

Figure 7: Causal graph of the CausalCircuit dataset.

ILCMs are again able to disentangle the causal variables reliably and better than the acausal baselines, see Tbl. 1. As shown in Appendix D, the slot attention model fails because the lights have no limited spatial extent and thus are not well represented by segments of the image. Interventions are identified with high accuracy. ILCMs also correctly learn the causal graph shown in Fig. 6, both when extracted with ENCO and with our heuristic algorithm. In Fig. 6 we demonstrate how ILCMs let us infer and manipulate causal factors and reason about interventions.

By studying variations of this dataset, we tested the limitations of our method. We find that it works reliably only as long as the causal variables are continuous (that is, when we model the lights with a continuous intensity). As soon as we consider discrete states, the assumptions of our identifiability theorem are violated and the model has difficulty disentangling these variables.

## 5.4 Scaling with graph size

Finally, we study how LCMs scale with the size of the causal system. We generate simple synthetic datasets with $\mathcal{X} = \mathcal{Z} = \mathbb{R}^n$. For each dimension $n$, we generate three datasets, using linear SCMs with random DAGs, in which each edge in a fixed topological order is sampled from a Bernoulli distribution with probability 0.5. The causal variables are mapped to the data space through a randomly sampled $SO(n)$ rotation.

We find that ILCMs are able to reliably disentangle the causal variables in systems with up to approximately 10 causal variables, see Fig. 8. In this regime, the true causal graphs are also identified with good accuracy, see Sec. D.5 of the supplementary material. In larger causal systems, both disentanglement and graph accuracy become worse; more work is required to improve the scaling of our approach to causal representation learning.

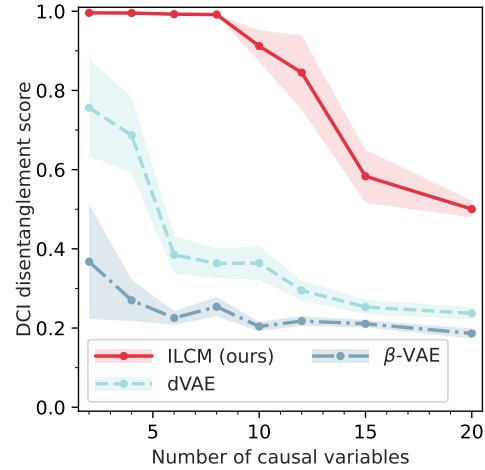

Figure 8: Scaling with graph size. LCMs disentangle causal variables robustly in simple systems with up to $\sim 10$ causal variables.

# 6 Discussion

What makes a variable causal? One school of thought is that it that causal variables are those aspects of a system that can be intervened upon [34]. Following this logic, we find it interesting to ask: can we uniquely determine the causal variables underlying a system just by observing the effect of interventions?

In this work we have found a partial answer to this question: we have shown in theory and practice that under certain assumptions, causal variables and their causal structure are identifiable from low-level representations like the pixels of a camera feed if the system is observed before and after random, unlabeled interventions. Our identifiability theorem extends the results by Locatello et al. [5] from independent factors of variation (trivial causal graphs) to arbitrary causal graphs.

Latent causal structure can be described in a variational autoencoder setup. However, a straightforward, explicit parameterization of the causal structure requires simultaneously learning the variables and the causal graph. We found that leads to challenging optimization problems, especially when scaling to larger systems. As a

Table 1: Experiment results. We compare our ILCM-E (using ENCO for graph inference) and ILCM-H (with a heuristic for graph inference) to disentanglement VAE (dVAE-E), unstructured $\beta$-VAE, and slot attention baselines. We show the DCI disentanglement score ($D$), the accuracy of intervention inference (Acc), and structural Hamming distance (SHD) between learned and true graph. Best results in bold.

| Dataset | Method | $D$ | Acc | SHD |
|---|---|---|---|---|
| 2D toy data | ILCM-E (ours) | **0.99** | **0.96** | **0.00** |
| | ILCM-H (ours) | **0.99** | **0.96** | **0.00** |
| | dVAE-E | 0.35 | **0.96** | 1.00 |
| | $\beta$-VAE | 0.52 | – | – |
| Causal3DIdent | ILCM-E (ours) | **0.99** | **0.98** | **0.00** |
| | ILCM-H (ours) | **0.99** | **0.98** | 0.17 |
| | dVAE-E | 0.82 | **0.98** | 1.67 |
| | $\beta$-VAE | 0.66 | – | – |
| | Slot attention | 0.60 | – | – |
| CausalCircuit | ILCM-E (ours) | **0.97** | **1.00** | **0.00** |
| | ILCM-H (ours) | **0.97** | **1.00** | **0.00** |
| | dVAE-E | 0.34 | **1.00** | 5.00 |
| | $\beta$-VAE | 0.39 | – | – |
| | Slot attention | 0.38 | – | – |

more robust alternative, we introduced implicit latent causal models (ILCMs), which parameterize causal structure without requiring an explicit graph representation. We also discussed two algorithms for extracting the learned causal mechanisms and graph after training.

In first experiments, we demonstrated that ILCMs let us reliably disentangle causal factors, identify causal graphs, and infer interventions from unstructured pixel data. For these experiments, we introduced the new CausalCircuit dataset, which consists of images of a robot arm interacting with connected switches and lights.

The setting we consider is motivated by a potentially useful scenario: learning causal structure from passive observations of an agent (or demonstrator) interacting with a causal system. However, it is currently far from practical. Our identifiability result relies on a number of assumptions, including that interventions are stochastic and perfect, that all atomic interventions may be observed, and that the causal variables are real-valued. In addition, realistic temporal sequence data are not likely to exactly correspond to a causal system before and after an intervention (while preserving the noise variables); whether our causal abstraction provides a useful approximation remains to be tested. We discuss these requirements and their potential relaxation in Appendix B. Similarly, our practical implementation has so far been restricted to simplified datasets with relatively few, continuous causal variables, and when trying to relax these limitations we saw the model performance decrease quickly. While more work will be required to make latent causal models applicable to real-world settings, we believe that our results demonstrate that causal representation learning is possible without explicit labels.

**Acknowledgments** We want to thank Joey Bose, Thomas Kipf, Dominik Neuenfeld, and Frank Rösler for useful discussions and Gabriele Cesa, Yang Yang, and Yunfan Zhang for helping with our experiments.

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
