# OpenReview forum: "Weakly supervised causal representation learning"
_NeurIPS.cc/2022/Conference — NeurIPS 2022 Accept_

### Official Review · Reviewer_HAYV · 2022-07-05

**Rating:** 7
**Confidence:** 4
**Soundness:** 4 excellent
**Presentation:** 4 excellent
**Contribution:** 4 excellent

**Summary:**

This paper provides identifiability results for causal discovery under a weakly supervised setting where we have access to data generated from perfect atomic interventions. Along with the theoretical results, it also proposes two algorithms for learning, one requiring an explicit DAG search and the other can learn the graph implicitly.


**Questions:**

As it stands, I think it is a very good paper and it should be accepted. A few questions and (minor) criticisms:

- This is the question most directly related to the current paper - I didn't find it clear in the paper as to why ILCM should perform better than ELCM. I can see that ILCM doesn't require explicit graph search which is nice, but can it not also be stuck in local minima just like ELCMs do, if it has found an (implicit) incorrect graph? Perhaps you could clarify this.

- The situation where we have interventional data from perfect and atomic interventions is not very applicable. Although the current work is a good starting point, further investigation into how to relax the interventional data assumption is needed.

**Limitations:**

The authors should be commended for calling out the limitation of their assumptions. I am quite interested in understanding what happens to the identifiability results if we still have data from heterogeneous regimes, but possibly from imperfect/non-atomic interventions? Can the results be extended? In the paper, the authors give a counter example for when the intervention is imperfect, but what about the case when we don't have atomic interventions?

**Strengths And Weaknesses:**

This is a nice result and I enjoyed reading the paper! The strength of the paper are:

- The theoretical result is novel and allows identifiability of causal graphs with interventional data. This is important and future algorithms can be made based on this.

- The ILCM description is nice as it allows learning of the graph without explicit graph searching.

- The experiments are non trivial and shows off the implication of the result.

- The paper is well presented and written clearly. It was an enjoyable read.

---

> ### Author Response · Authors · 2022-08-02
> **Response to reviewer HAYV**
>
> Thank you for the helpful review and the encouraging words. In the following, we would like to address your two main questions and comments.
>
> **Advantage of ILCMs over ELCMs**: It is a good question why exactly ILCMs perform so much better than ELCMs. You are right, both models can initially learn a wrong graph orientation. However, we believe that it is much easier for an implicit LCM to "unstuck" itself: the neural solution functions in the ILCM can intermittently take on configurations that do not correspond to a valid DAG, for instance with $e\_2$ influencing the distribution of $\tilde{e}\_1$ and ${e}\_1$ influencing the distribution of $\tilde{e}\_2$. This allows the ILCM to smoothly transition out of a bad graph configuration into the correct graph without driving up any loss terms. In the end, the learned solution functions always correspond to valid DAGs, which can for instance be extracted with the heuristic algorithm we proposed.
>
> **Applicability of assumptions**: We agree that our work is far from applicable to real-life problems and that our identifiability theorem (and the practical implementation) rely on strong assumptions (in particular that of perfect interventions) that will often not be satisfied. We are glad to hear that you share our opinion that our work is nevertheless a useful starting point, and agree that there should be more research into relaxing these assumptions. In Section B of the supplementary material, we attempt to summarize our assumptions as well as some thoughts on which of the assumptions are likely to be relaxed in future work.
>
> **Updated paper**: We have updated our submission with four main changes. In the PDF file, we have highlighted text changes in green.
>
> 1. To study whether our method scales to higher-dimensional data spaces, we have updated our CausalCircuit dataset to 512x512 resolution. ILCMs are still able to disentangle causal factors (DCI disentanglement of 0.97) and infer the true causal graph, outperforming the acausal baselines.
> 2. To study scaling to more complex causal systems, we have added a new experiment with simple synthetic datasets with between 2 and 20 causal variables and random DAGs. ILCMs scale to around 10 causal variables: in this regime they enable disentanglement scores close to 1 and outperform the baseline in terms of the accuracy of the inferred causal graph. Scaling ILCMs to 15 or more variables will require more research.
> 3. In addition to the DCI disentanglement score, we now report the completeness and informativeness scores proposed in Eastwood & Williams, ICLR 2018, which support the same conclusions as the disentanglement score.
> 4. We have improved the text in various places.
>
> Thank you again for your review and comments.

---

### Official Review · Reviewer_eczw · 2022-07-09

**Rating:** 4
**Confidence:** 4
**Soundness:** 2 fair
**Presentation:** 3 good
**Contribution:** 2 fair

**Summary:**

This paper has contribution in two parts: Identifiability of causal models of high level (latent) variables from low-level data like that of pixels of images based on weak supervision, and two practical algorithms to learn causal models based on the weak supervision. The identifiability result is based on the assumption that pairs of datum where one of them is not intervened upon and the other is, after an atomic hard intervention to one of the high level variables, is available for training. The latent variables need to be continuous for the identifiability result to hold.
In addition, there are two algorithms, Explicit Latent Causal Models which are based on explicit modelling of causal graphs as well as Implicit Latent Causal Models which do not model the causal graph directly but rather parameterise the noise encodings. Experiments are performed on very simple datasets to demonstrate the practical algorithms.

**Questions:**

Here are some of the questions I have for the authors:

- For Causal3Dindent dataset, only 6 graphs are considered, why not more? And are the considered six graphs random? Also please give the confidence intervals.

- I do not get the intuition behind using a beta term for the ELBO. Is it not the case that the goal of the current setup is to get the correct causal graph? Encouraging disentanglement through the beta term might be encouraging more independent factors, however, that might not be necessarily the case with hierarchical latent spaces with causal graphs.

- Does the ILCM also suffer from optimisation issues in higher dimensions, and if so, what advantages then practically speaking does it offer as against ELCM.



**Limitations:**

One of the main limitations of the proposed approach is that the causal variables are assumed to be causally sufficient (i.e. they are apriori known or given in some way through slots etc and they alone define a causal model). However, this assumption is notoriously hard to be true in any realistic dataset. For any image, the cause of a particular variable could be outside the image itself, and hence hidden. This point should be discussed and highlighted. In my opinion, a true practical algorithm should be able to handle latent variables in causal models as well. Nevertheless, I feel that this aspect should be discussed as it is an important aspect of causal representation learning.

**Strengths And Weaknesses:**

**Strengths**:

The identifiability result follows from the work of Locatello et al. 2020 where the identifiability of disentangled representations are presented. The authors in this paper do a good job of extending it to incorporate the notion of graphs in latent variable models and hence it enables the identifiability of arbitrary causal models from unknown interventions.

In addition, the idea of implicitly parameterising the causal model and then using the latent variable data to learn a causal model from an off-the-shelf algorithm is very interesting as this is something I believe has not been done before in causal representation learning.

**Weaknesses**:

The identifiability result is based on some assumptions which are hard to verify. For any given dataset of samples, it is hard to verify, let alone know, if the data samples were produced from a single hard intervention on one of its latent variables or if there was a soft intervention or a multi-target intervention. So in this regard, I am a bit unsure in which real world image data does these assumptions hold. The presented results are on very simple and hand-designed datasets where the assumptions are made to hold. Also see below in limitations regarding other assumptions which are not addressed.

The main weakness of the paper is that the experiments are very weak, to say the least. The datasets are already simple in the sense that all the assumptions of the identifiability result are respected, and moreover, the number of high level variables on which the causal graph is defined is utmost 4, which is a very small number. For reference, for up to 4 variables, all the DAGs can even be enumerated and is a much simpler task. It is disappointing to see that even the implicit LCM algorithm which does not model a causal graph explicitly is not shown to work on more number of variables (10 or 20 for example). For the ELCM, the authors mention that it takes a performance hit when the number of variables increases due to optimisation issues. If it is the same case with ILCM as well, then I would argue that the presented algorithms are far from being practical. This could be a case of identifiability does not imply learnability, unless the authors convincingly demonstrate in higher dimensions that this works reasonably.

Adding to the above point, I suggest the authors to consider a synthetic dataset (maybe with a random nonlinear SCM plus normalizing flow like in the pedagogical experiment) where they can create data samples such that number of high level variables are 10 (or more). Multiple such random datasets could be produced and trained on with an ILCM. This would at least highlight how the ILCM performs in higher dimensions. Given that ENCO can handle variables upto 1000 variables, I do not see any limitations in performing these experiments.

The proposed algorithm achieves both high disentanglement scores as well as SHD. But this seems a bit counterintuitive. Disentanglement measures how much each of the factors of variation are disentangled from each other while SHD measures how well the ground truth graph is recovered. If the ground truth graph is not extremely sparse, then both cannot be high. Maybe this is only a behaviour in low dimensions which the authors consider. In addition, it could be that DCI score maybe captures something similar to a structural metric. I personally do not think that it is important to measure disentanglement if the focus is to get the correct graph. If it is still important for some tasks, I encourage the authors to measure other disentanglement metrics as well.

In summary, I feel that the theoretical contribution is very interesting and can be a stepping stone for identifiability results with more relaxed assumptions. However, given that the experiments are weak and the practical algorithms need more realistic settings, I think that more work needs to be done in this aspect before the paper can be accepted.

---

> ### Author Response · Authors · 2022-08-02
> **Response to reviewer eczw**
>
> Thank you for your review and the detailed comments.
>
> **Assumptions**: We agree that it is hard to verify if the assumptions of our identifiability theory hold in any given dataset. In most real-life datasets, they will likely not hold. We wish that we already had a causal representation learning algorithm that works in realistic settings, but unfortunately, this problem is far from being solved. Other recent works on causal representation learning require similar or stronger assumptions, see for instance CausalVAE (M. Yang et al, CVPR 2021) or CITRIS (P. Lippe et al, ICML 2022). We hope that we are transparent about these limitations, which we discuss in our conclusions and in Section C in the supplementary material.
>
> Despite being still some distance away from realistic use cases, we believe that our work contributes to better understanding what kind of information can make causal structure identifiable. We are, to the best of our knowledge, the first to show even in principle that causal variables and arbitrary causal graphs can be identified from pairs of pre- and post-intervention data. We believe that this result may be valuable to guide further progress in the field.
>
> **Experiments with larger graphs**: Thanks for the feedback on our experiments. We would like to stress that the goal of these experiments was not to study realistic settings, but to verify that our identifiability result translates into a learning algorithm that can under certain assumptions identify causal variables and graphs.
>
> We added an experiment to study scaling to larger causal graphs. Thank you for pointing out this omission in our original submission and suggesting this experiment. We generate simple synthetic datasets with linear SCMs, random DAGs, and SO(n) decoders with 2 to 20 causal variables. We find that ILCMs scale to around 10 causal variables, visible for instance in these mean disentanglement scores:
>
> | Causal variables | ILCM | dVAE | β-VAE |
> |---:|---:|---:|---:|
> | 8 | 0.99 | 0.36 | 0.25 |
> | 10 | 0.91 | 0.36 | 0.20 |
> | 12 | 0.85 | 0.29 | 0.22 |
> | 15 | 0.58 | 0.25 | 0.21 |
>
> For more results and error bars see Fig. 6 in the updated paper, for graph inference results Fig. 13 in the updated supplementary material. Scaling ILCMs to systems of 15 variables or more will require more research.
>
>
> **Higher-dimensional data**: We explore the scaling with data dimensionality by updating the CausalCircuit dataset to a resolution of 512x512 pixels. ILCMs are still able to learn the correct causal variables (DCI disentanglement of 0.97) and recover the true causal graph.
>
> **Disentanglement**: We believe there may be a misunderstanding due to the fact that the word "disentanglement" is used to mean two different things: (i) that a set of latent variables is in a one-to-one correspondence with a set of ground-truth factors, or (ii) that a set of latent variables follows a distribution such that each pair of latent variables are independent.
>
> The disentanglement score we report measures (i). While this metric is often used in the setting where variables are disentangled based on property (ii), this is not necessary for the DCI metrics to be meaningful. They can still measure disentanglement (in sense (i)) between correlated variables, like our causal variables in our case; see for instance, Träuble et al, "On disentangled representations learned from correlated data" (ICML 2021).
>
> **Questions**:
>
> > For Causal3Dindent dataset, only 6 graphs are considered.
>
> For 3 nodes, there are 6 unique DAGs (up to a permutation of the nodes). We created one dataset for each of these DAGs. The mapping from high-level concepts to the nodes in these graphs is random, as are the causal mechanisms.
>
> > Does ILCM also suffer from optimisation issues in higher dimensions?
>
> In our experiments, ILCMs proved robust to train. Unlike ELCMs, ILCMs did not require scanning over multiple random seeds to learn the correct variables and graphs. We attribute this to ILCMs being less prone to local minima in the loss landscape from wrongly oriented graph edges. ILCMs (but not ELCMs) are able to escape from such configurations without incurring a loss penalty because they can intermittently take on non-acyclic configurations.
>
> **Updated paper**: We have updated our submission with four main changes, highlighted in green in the PDF file:
>
> 1. To study scaling to higher-dimensional data, we have updated our CausalCircuit dataset to 512x512 resolution.
> 2. To study scaling to more complex causal systems, we have added a new experiment with synthetic datasets with between 2 and 20 causal variables.
> 3. In addition to disentanglement, we report the completeness and informativeness scores proposed in Eastwood & Williams, ICLR 2018.
> 4. We have improved the text in various places.
>
> Thank you again for your comments and suggestions. They were very helpful in improving our manuscript.

---

### Official Review · Reviewer_UyEP · 2022-07-10

**Rating:** 4
**Confidence:** 3
**Soundness:** 3 good
**Presentation:** 1 poor
**Contribution:** 3 good

**Summary:**

This paper aims to identify the latent causal models from the observational data, where data can be intervened by some pre-defined distributions. To achieve this goal, the authors firstly demonstrate that if the observation data is generated according to equation 1, then the causal model can be identified. Based on this theory, the authors design two specific models to learn the causal variable and causal structure. The first model is based on VAE and the second one infers the intervention from the noisy variable. In addition, the authors also discuss the potential functions of their proposed models. In the experiments, the authors demonstrate the effectiveness of their model on the tasks of pedagogy, Causal3DIdent and CausalCircuit.

**Questions:**

See the Strengths And Weaknesses

**Limitations:**

The authors have discussed the limitations of their model.

**Strengths And Weaknesses:**


In general, I believe the studied problem is interesting. However, I find the paper quite hard to read. There are so many contents referred to the appendix or other papers. For example, the formal definition of diffeomorphic function, at least, there should be some intuitive explanations. What is faithful in the causal domain? What is p_M^X? Ideally, the reader should clearly understand what the authors would like to deliver. However, I think there are a lot of concepts that should not be regarded as common knowledge (it is not necessary for the reviewers to read the appendix). Because of these unclear notations, I have trouble in reading this paper.

There should be an explicit algorithm on how to train the ELCM and ILCMs. I cannot capture how to relate the solution of ELCM and ILCMs with the theoretical results in section 3. Whether the results learned from ELCM and ILCMs can lead to a optimal result of theory 1?

I'm wondering how accurate are the inferred interventions, and how the error of the inference influence the following causal prediction? Whether there are some strategies to overcome the inference error?

Considering that the paper seems to study a very interesting problem, I tend to be slightly positive. However, I can lower my rating if the other reviewers do not support this paper.

---

> ### Author Response · Authors · 2022-08-02
> **Response to reviewer UyEP**
>
> Thank you for your review and helpful, constructive feedback. Here we will address your main points.
>
> **Presentation**: We wholeheartedly agree that the space constraints made it impossible to explain the underlying concepts, our theory, and our practical implementation and experiments in a lot of detail. We are determined to use the space we have as well as we can.
>
> Thank you for pointing out two omissions where we failed to define important concepts. We have added short explanations of diffeomorphisms and faithfulness in our updated version of the paper and also added a reference that explains common concepts in causality in more depth. We also improved the text in a number of other places and hope that these changes improved readability. We would be grateful if you could point us to any other parts of the paper that are particularly hard to follow.
>
> > What is faithful in the causal domain?
>
> Loosely speaking, a causal model is faithful if there are no accidental independence relations in the data: there are no variables that are causally connected according to the causal graph, but are independent in the data distribution. This is a very common assumption in causal discovery (see e.g. Hyttinen et al., "Experiment Selection for Causal Discovery", JMLR 2013), since without faithfulness it is often impossible to identify all edges in the causal graph.
>
> > What is $p_M^X$?
>
> This is the data distribution in the weakly supervised setting (pre- and post-intervention data) according to an LCM M. We define it in Definition 3.
>
> **Training algorithm**: We have added an explicit algorithm to Section C.2 of the supplementary material that shows the complete training procedure for ILCMs. ELCM training is very similar, the differences are explained in Section E in the supplementary material.
>
> **Accuracy of intervention inference**: We report the accuracy of intervention inference in Table 1 in the "Acc" column. We find that our methods are consistently able to classify interventions with an accuracy of over 95\%. The same is true for the dVAE baseline, but despite the good intervention accuracy, that approach is not able to disentangle the causal factors and learn the causal graph correctly.
>
> Indeed, we find that it is crucial to accurately infer interventions in order to learn disentangled representations and correct causal graphs. In earlier experiments where the intervention inference failed, the model was never able to disentangle the causal variables.
>
> You raise the interesting question of how to overcome inference errors. We developed a training schedule that improved the quality of intervention inference. Early in training, we just train the noise encoder and decoder as well as the intervention encoder (and set the solution distributions to a uniform probability density that is not trained yet). This substantially stabilizes the training of the intervention encoder, as it avoids contributions to the loss from the randomly initialized solution functions. We describe this schedule in Section C.2 of the supplementary material.
>
> **Updated paper**: All in all, we have updated our submission with four main changes. In the PDF file, we have highlighted text changes in green.
>
> 1. To study whether our method scales to higher-dimensional data spaces, we have updated our CausalCircuit dataset to 512x512 resolution. ILCMs are still able to disentangle causal factors (DCI disentanglement of 0.97) and infer the true causal graph, outperforming the acausal baselines.
> 2. To study scaling to more complex causal systems, we have added a new experiment with simple synthetic datasets with between 2 and 20 causal variables and random DAGs. ILCMs scale to around 10 causal variables: in this regime they enable disentanglement scores close to 1 and outperform the baseline in terms of the accuracy of the inferred causal graph. Scaling ILCMs to 15 or more variables will require more research.
> 3. In addition to the DCI disentanglement score, we now report the completeness and informativeness scores proposed in Eastwood & Williams, ICLR 2018, which support the same conclusions as the disentanglement score.
> 4. We have improved the text in various places. Thank you for your helpful suggestions for this.
>
> Thank you again for your review and comments, which helped us improve our manuscript.

---

### Official Review · Reviewer_GuNY · 2022-07-10

**Rating:** 7
**Confidence:** 4
**Soundness:** 3 good
**Presentation:** 3 good
**Contribution:** 3 good

**Summary:**

### Problem setting
The paper addresses the causal representation learning task, i.e., inferring high-level causal latent variables from low-level observations. Specifically, it considers a weakly-supervised setting in which pairs $(x,x’)$ of pre- and post-intervention observations are available. This can be seen as a generalization of Locatello et al. [5] to a setting in which the latents are not mutually independent but causally related. The paper formalizes this setting via latent causal models (LCMs) which consist of a structural causal model (SCM) over latent variables $z$ and a decoder $g$ mapping $z$ to observations $x$ (as also considered in previous works).

### Theory
The main theoretical contribution (Thm. 1) is an identifiability result stating that, given an infinite number of pairs $(x,x’)$ resulting from all perfect stochastic single-node interventions (those that set a causal variable $z_i$ to a new noise variable, thus completely removing any influence from its causal parents), the true LCM is identified up to a permutation and element-wise invertible reparametrisation of the causal variables.

### Algorithm/Method
Based on this insight, the paper then proceeds to investigate algorithmic approaches to learning such LCMs from data. Specifically, two VAE-based approaches are investigated which either represent the causal variables and graph explicitly or only implicitly (ELCMs and ILCMs, respectively). ELCMs are only briefly discussed at a high level and are dismissed due optimization challenges. ILCMs, in which the latent space corresponds to the exogenous noise variables in the LCM, are portrayed as a more promising practical approach. ILCMs consist of an intervention encoder, a noise encoder and decoder, and a solution function (a map between exogenous noise and endogenous causal variables) and are trained by maximizing the corresponding ELBO. Two methods are proposed for post-hoc extracting causal structure from a trained ILCM.

### Experiments
In experiments on three synthetic and image datasets, ILCMs are compared with other VAE architectures and SlotAttention w.r.t. DCI score, accuracy in inferring intervention targets, and structural Hamming distance and are shown to compare favourably. A new Mujoco-based dataset called CausalCircuit is also introduced in the process.

**Questions:**

### Main questions and comments
My main concerns and questions relate (i) to the concept of representing SCMs through the combination of exogenous variables and a solution function (which is also known as the reduced form SCM in the literature), as well as to (ii) the soundness of evaluation based on DCI scores.

(i) To my understanding, this representation of SCMs is not equivalent to the original SCM, but strictly less expressive in that the reduced form SCM cannot model arbitrary interventions to the true causal mechanisms---see, e.g., Sec. 10 of (Schölkopf, B. & von Kügelgen, J. From Statistical to Causal Learning. 2022). Would the authors agree with this statement?

I would like to better understand how the new solution functions $\tilde{s}_I$ are defined:
- Does this not require knowledge of the true mechanisms?
- How does this relate to the assumption of perfect, stochastic interventions?
- What aspect of the theory and proposed method would break down, e.g., for hard interventions (which set $z_i$ to a constant) or for soft interventions that preserve some dependence on the parents?

Some more specific questions regarding the solution function(s):
- Before Defn. 1, the new mechanism $\tilde{f}_i$ is defined over the same exogenous domain $\mathcal{E}_i$, but in Defn. 3, $\tilde{\epsilon}_i$ seems to be a new noise variable with different domain. Can you clarify?
- Regarding footnote 3, do you have a formal argument or reference to back up this claim?
- In Sec. 4.2, paragraph “Latents”, the noise variables are defined as $e=s^{-1}(z)$ and $\tilde{e}=s^{-1}(\tilde{z})$ and it is correctly stated that the latter corresponds to noise value that would have generated  $\tilde{z}$ under the *unintervened* SCM mechanisms. In the next paragraph, it is then stated that only those components of $\epsilon$ corresponding to intervened nodes change. Where exactly in Appendix A is this proven?  While I found this result surprising at first, after looking at some toy examples, it seems to hold. However, the way in which $\epsilon_I$ needs to change will depend on the value of the other noise terms $\epsilon_{-I}$, thus rendering them dependent. Could you comment on the relevance of this?

(ii) Based on my understanding, the DCI score of Eastwood and Williams is tailored to a setting with mutually independent ground truth factors since it is based on predicting different ground truth latents from the learnt representation. It would seem that this is no longer necessarily a sound evaluation when the $z_i$ are dependent.
- Did you consider this aspect?
- What was your choice for the feature importance matrix?
- Do you report only the disentanglement (D) score or the whole DCI score (average of D, completeness C and informativeness I)? Since the abbreviation D was used, this was not completely clear.

### Other more minor comments, questions, and suggestions
- l.4: “this requires…” makes it sound like a necessary condition, whereas what is shown is that the considered setting is sufficient. Consider rewording, e.g., “this involves…”
- the restriction to perfect stochastic interventions seems important enough to be mentioned in the abstract and introduction
- Fig. 1: consider adding an explanation of what orange nodes represent (changed post-intervention values?)
- Related work: I would consider adding the following causal representation learning references:
- - Adams, J., Hansen, N., & Zhang, K. Identification of partially observed linear causal models: Graphical conditions for the non-gaussian and heterogeneous cases. Advances in Neural Information Processing Systems 34, 2021.
- - Xie, F., Cai, R., Huang, B., Glymour, C., Hao, Z., & Zhang, K. Generalized independent noise condition for estimating latent variable causal graphs. Advances in Neural Information Processing Systems 33, 2020.
- - Kivva, B., Rajendran, G., Ravikumar, P., & Aragam, B. Learning latent causal graphs via mixture oracles. Advances in Neural Information Processing Systems 34, 2021.
- - Chalupka, K., Perona, P., & Eberhardt, F. Visual causal feature learning. Uncertainty in Artificial Intelligence, 2015.
- - Beckers, S., & Halpern, J. Y. Abstracting causal models. AAAI Conference on Artificial Intelligence, 2019.
- l.104: including a intervention distribution in defining SCMs relates to (Rubenstein, P., Weichwald, S., Bongers, S., Mooij, J., Janzing, D., Grosse-Wentrup, M., & Schölkopf, B. Causal Consistency of Structural Equation Models. Uncertainty in Artificial Intelligence, 2017.)
- Defn. 2: What does “compatible” mean here? What implications does this have for the (obs./int./cf.) distributions implied by the two SCMs, will they match?
- Thm. 1: consider adding “in the sense of Defn. 2” to statement 2.
- l. 151: AFAIK, Pearl defines hard interventions as those which set a variable to a constant; in this sense, the considered stochastic interventions are not hard (in general)
- Sec. 4.1: some more details on ELCMs (e.g., form of the prior $p(z,z’)$) would be helpful
- Sec. 4.2: the idea of ILCMs where structure is embedded implicitly seems related to: Leeb, F., Lanzillotta, G., Annadani, Y., Besserve, M., Bauer, S., & Schölkopf, B. Structure by architecture: Disentangled representations without regularization. 2020.
- l. 214: what if the true post-intervention distribution is non-Gaussian? Why is Gaussianity needed here, or could a more flexible density also be used? I understand this is only the base density used to encode a more complex distribution over the noise variable, but this seems somewhat counterintuitive: do we not mostly care about the distribution of causal variables in the end? In this sense, forcing it to be Gaussian seems restrictive.
- L. 215 ff.: I was not able to really follow this and the next paragraph in satisfying detail; perhaps consider expanding/clarifying.
- L. 238: since $q(I|x, x’)$ involves $\mu_e$, does this not depend on the noise encoder? Is $\mu_e$ shared between both?
- L. 296: is the graph of dVAE-E not always the empty graph by construction?
- L. 300: are objects/latents learned by SlotAttention assumed/constructed to be independent?
- Evaluation: it could be interesting to also evaluate how well different approaches capture the true interventional distribution, e.g., by looking at an average KL between different true and inferred interventional distributions.
- Fig.5: which ground truth variables are intervened upon here? Also, some more details on how the ILCM intervention is generated exactly would be helpful.
- L.369: any intuition on what the failure mode is here?
- (L.378) Generally, I think the pre- and post-intervention terminology is slightly misleading: Typically, interventions are associated with layer 2 queries in which the exogenous variables are not shared but randomly resampled. In the considered setting, the exogenous variables are shared which makes it a layer 3 / counterfactual query. This point has been made in a similar multi-view/weakly-supervised setting by von Kügelgen et al. [14, Sec. 3] and I think this could be articulated more clearly throughout. In particular, “pre-“ and “post-“ suggest a temporal succession, which is not the same as the considered *hypothetical* intervention under the same context/background condition/noise values; the analogy to agents observing the effect of actions may thus only hold approximately, as there some noise variables may change (naturally, i.e., not as the result of the intervention).

**Strengths And Weaknesses:**

## Post-Rebuttal Update
After discussion with the authors, some of my doubts and questions (particularly regarding the use of solution functions) have been resolved and addressed, and I have decided to increase my score as a result. I believe this is a solid paper that would benefit the NeurIPS community and recommend acceptance.

I encourage the authors to take serious the suggestions brought up during the review phase to further improve the paper, particularly its accessibility and presentation, as they already indicated they would.
____

*Disclaimer / review context: I have read the main paper carefully, but only skimmed the Appendix. Due to lack of familiarity with category theory, I did not check the proof of the main theorem and can thus not judge its soundness. I am quite familiar with the setting and related literature, but was not able to follow certain aspects in detail---specifically, the exact use of different solution functions---which I hope to be clarified during the discussion period. My score reflects my current impression of the paper, but I remain open to adjusting my score based on the authors' response.*

### Strengths
- The paper is generally well-written and well-motivated.
- The problem setting of learning causal representation from weak supervision is novel and highly relevant.
- The paper makes contributions to both theory (identifiability result) and algorithms (ELCMs and ILCMs) for causal representation learning.
- The paper is well positioned in the related literature.
- The paper is honest and transparent in discussing limitations, both regarding theoretical assumptions and implementation.
- The paper introduces useful concepts such as isomorphisms between LCMs, which are of interest to the field of causal representation learning beyond the present work.

### Weaknesses
- Due to addressing both theory and algorithmic approaches, many technical details are deferred to the Appendix, which makes it hard to understand the proposed method in sufficient detail based on just the main text.
- I have some doubts/reservations regarding the soundness of the proposed ILCMs (specifically, regarding the representation of SCMs through the noise variables and solution functions) and the evalution in terms of the DCI score (see below for more details).

---

> ### Author Response · Authors · 2022-08-02
> **Response to reviewer GuNY (1 of 2)**
>
> Thank you for your thorough review and the many helpful suggestions. In this response, we focus on your main points, but take all the suggestions into account in our revised paper.
>
> **Points regarding solution functions**:
> > [The reduced SCM representation] is [...] strictly less expressive in that [it] cannot model arbitrary interventions
>
> Due to our assumption that the causal mechanisms are bijective functions from the noise and that the graph is acyclic, the causal mechanisms can be recovered from the solution. Consider a simple two variable case, with graph $A \to B$ and mechanisms $z\_A=f\_A(\epsilon\_A), z\_B=f\_B(z\_A, \epsilon\_B)$. Then the solution is $s(\epsilon\_A, \epsilon\_B)=(f\_A(\epsilon\_A), f\_B(f\_A(\epsilon\_A), \epsilon\_B))$. From this function, we can recover the mechanism: $f\_A(\epsilon\_A)=s(\epsilon\_A, ...)\_A, f\_B(z\_A, \epsilon\_B)=s(s^{-1}(z\_A, ...)\_A, \epsilon\_B)\_B$. Note that the inverse is well-defined, because the $A$ component of $s$ is non-constant only in the first argument $\epsilon\_A$, and is a bijective function thereof by assumption. This process requires us to know a topological ordering of the DAG, which can also be recovered from the solution: it is any permutation of rows and columns so that the Jacobian $\partial s\_i / \partial \epsilon\_j$ is triangular on all inputs $\epsilon$. This argument generalizes to any DAG.
>
> So including the bijectivity assumption, we disagree that the reduced SCM contains less information than the original SCM.
> In the ILCM, to model any perfect intervention on variable $i$, we encode $x$ to $e$, sample (for stochastic interventions) or set (for hard interventions) $\tilde z\_i$, transform with $s_i^{-1}$ to $\tilde e$ and decode.
>
> We have update the description of the ICLM in the paper and appendix C.3 in the supplementary material to clarify these points.
>
> > How are intervened solution functions  $\tilde{s}\_I$ defined?
>
> In the ELCM, for each intervened mechanism $i \in I$, we have an intervened causal mechanism $\tilde f\_i : \mathcal{E}\_i \to \mathcal{Z}\_i$, mapping a noise variable to the stochastic intervened variable independent of its original parents. For any $j \not \in I$, we have the original causal mechanism  $\tilde f\_j=f\_j$. The solution $\tilde s\_I$ then follows from the intervened mechanisms $\tilde f$ by recursive substitution, just as with the original solution.
>
> > Does this not require knowledge of the true mechanisms?
>
> No, we prove identifiability from just observing the $p(x, \tilde x)$.
>
>
> > What aspect of the theory and proposed method would break down, e.g., for hard interventions (which set $z\_i$ to a constant) or for soft interventions that preserve some dependence on the parents?
>
> As described above, our learned model can perform perfect, non-stochastic interventions. However, to learn it, we are using unknown interventions sampled from some distribution. If we would train with only a single intervention value for each variable, the intervened distribution $\tilde z$ would no longer have full support and the solution function not a diffeomorphism and our method would break down.
>
> Please see appendix B for a discussion on generalization to soft interventions: that case is not identifiable.
>
> > Domain of intervened noise variable
>
> The domain for each variable is just $\mathbb{R}$, so those spaces are equal. The subscript $\tilde {\mathcal{E}}\_i$ denotes that this is the distribution over the interventional noise $\tilde \epsilon\_i$.
>
> > footnote 3
>
> Which footnote do you mean? Our initial version does not contain a footnote 3.
>
> > Proof that only intervened noise variables change
>
> This is an excellent point, thank you very much. We removed this proof from an earlier version, as we don't rely on it in our main theorem anymore. For the ILCM construction, we still use it. We added it back to appendix C.
>
>
> **Suitability of DCI scores**:
>
> It is common, but not necessary to use DCI metrics for independent variables. The DCI metrics quantifies how much ground-truth factors and learned latents are in a one-to-one correspondence, but does not make assumptions about the joint distribution of latent and ground-truth factors. It is therefore also suited to measure disentanglement (in the sense of one-to-one correspondence between true factors and learned latents) between correlated variables, like the causal variables in our case.
>
> As an example, Träuble et al's "On disentangled representations learned from correlated data" (ICML 2021) also measures disentanglement with the DCI disentanglement score when the true factors are not independent.
>
> As for the implementation of the DCI metrics, we use gradient boosted trees (sklearn's implementation with default parameters) to construct the feature importance matrix. We added the completeness C and informativeness I scores to the results in the appendix. C and D give the same conclusions, while I is low for all models. Thank you for the suggestion.

---

> > ### Author Response · Authors · 2022-08-02
> > **Response to reviewer GuNY (2 of 2)**
> >
> > **Various questions**:
> >
> > > Defn. 2: What does “compatible” mean here? What implications does this have for the (obs./int./cf.) distributions implied by the two SCMs, will they match?
> >
> > Loosely speaking, ``compatible'' means that when you take all components of the first LCM and transform them under a permutation of the variables and elementwise transformations of each variable, you get out the second LCM. We make this statement more precise in Definitions 6, 8, and 9 in the appendix. There we phrase the idea of compatibility mostly through commutation relations like that shown in Eq. (5).
> >
> > > what if the true post-intervention distribution is non-Gaussian? Why is Gaussianity needed here, or could a more flexible density also be used?
> >
> > We only show identifiability of causal variables *up to elementwise reparameterizations of the causal variables* (and we think it is unlikely that a stronger identifiability statement could hold without explicit labels on the causal variables or distributional assumptions). Forcing each causal variable to follow a particular base distribution when intervened upon exactly resolves this ambiguity. We chose a Gaussian distribution for simplicity, but more complex distributions are certainly also possible. Note that any distribution on $\mathbb{R}$ with a smooth density that is non-zero everywhere is isomorphic to the standard Gaussian. This is a corollary of Lemma 1.
> >
> > > L. 238: since $q(I|x,x')$ involves $\mu\_e(x)$, does this not depend on the noise encoder? Is $\mu\_e(x)$ shared between both?
> >
> > Indeed, the noise mean function $\mu\_e(x)$ is shared between both. (Maybe you are wondering why we did not just write $q(I|e,e')$ then. That would be problematic because the distribution of the noise encodings depends on the intervention targets as $e, e' \sim q(x, x', I)$, and we needed to avoid cyclical dependencies. However, the mean function $\mu\_e(x)$ does not depend on I, so we are free to use that in $q(I|x,x')$.)
> >
> > > L. 296: is the graph of dVAE-E not always the empty graph by construction?
> >
> > Essentially yes, but there is some fineprint. The prior of the dVAE model is independent between the causal variables. However, it is still possible that the distribution of latents given by the data distribution pushed through the encoder (which can in general be different from the prior, see for instance I. Tolstikhin et al, "Wasserstein Auto-Encoders", ICLR 2018) is not independent. That is why ENCO occasionally detects edges in this distribution.
> >
> > > L. 300: are objects/latents learned by SlotAttention assumed/constructed to be independent?
> >
> > No, the slot attention baseline does not assume independence of latents, because it is not a probabilistic model. It is just a feed-forward neural network that is trained on pixel reconstruction loss.
> >
> > > L.369: any intuition on what the failure mode is here?
> >
> > With discrete ground-truth causal variables and continuous latent spaces in the VAE, the encoder is free to collapse the whole dataset into lower-dimensional manifolds in the latent space, even a single line, and to model all data pairs as an intervention along this one line.
> >
> >
> > **Updated paper**: We have updated our submission with four main changes, highlighted in green in the PDF file:
> >
> > 1. To study whether our method scales to higher-dimensional data spaces, we have updated our CausalCircuit dataset to 512x512 resolution. ILCMs are still able to disentangle causal factors (DCI disentanglement of 0.97) and infer the true causal graph, outperforming the acausal baselines.
> > 2. To study scaling to more complex causal systems, we have added a new experiment with simple synthetic datasets with between 2 and 20 causal variables and random DAGs. ILCMs scale to around 10 causal variables: in this regime they enable disentanglement scores close to 1 and outperform the baseline in terms of the accuracy of the inferred causal graph. Scaling ILCMs to 15 or more variables will require more research.
> > 3. In addition to the DCI disentanglement score, we now report the completeness and informativeness scores.
> > 4. We have improved the text in various places. Thank you for your helpful suggestions for this.
> >
> > Thank you again for your review and comments, which were very helpful in improving our submission.

---

> > > ### Comment · Reviewer_GuNY · 2022-08-09
> > > **Thanks for the response; continued discussion.**
> > >
> > > I thank the authors for their extensive response. I will update my review and respond to it separately.
> > >
> > > For now, since I still have some questions as to how exactly the solution function is used, I would like to use the remaining discussion time to ask the authors to further elaborate on the following concrete example:
> > >
> > > Consider an acyclic Markovian linear SCM with three variables:
> > > $$z_1:=\epsilon_1$$
> > > $$z_2:=z_1+\epsilon_2$$
> > > $$z_3:=z_1+z_2+\epsilon_3$$
> > >
> > > Suppose we observe $z=(1,2,4)$.
> > >
> > > Abduction yields $\epsilon=(1,1,1)$.
> > >
> > > We want to reason about the counterfactual under $do(z_2=0)$.
> > >
> > > We update the second equation, substitute $z_2=0$ into the third, and obtain the counterfactual (post-intervention) view $z=(1,0, 2)$.
> > >
> > > Now consider carrying out the same calculation from the reduced form SCM:
> > > $$z_1=\epsilon_1$$
> > > $$z_2=\epsilon_1+\epsilon_2$$
> > > $$z_3=2\epsilon_1+\epsilon_2+\epsilon_3$$
> > >
> > > Similarly, abduction on $z=(1,2,4)$ yields $\epsilon=(1,1,1)$.
> > >
> > > Now my main question is: how exactly do you calculate the counterfactual under $do(z_2=0)$ to arrive at the correct solution $z=(1,0, 2)$?
> > >
> > > Clearly, the naive approach of only replacing the second reduced form equation would have no effect on $z_3$ since $z_2$ (what has been actually changed) does not appear there, but is represented as $\epsilon_1+\epsilon_2$ (which are assumed constant for counterfactual reasoning). This is the issue I was referring to with "the reduced form SCM contains strictly less information": when used naively (without any modifications) it does not allow modelling interventions on the endogenous variables because their relationships are fully explained in terms of a single mapping from the exogenous variables.
> > >
> > > [Fixing this would likely require knowing exactly what the dependence of $z_2$ on  $\epsilon_1$ and $\epsilon_2$ is; this is what my comment "does this not require knowledge of the true mechanism?" referred to---I understand your result only uses information on $p(x,x')$.]
> > >
> > > Now based on your response, I understand (partially) that you propose some intermediate step as a fix which recursively constructs an alternative solution function (reduced form SCM) to model the intervention. Could you please explain and showcase this for the above example? I didn't fully get it and feel it might be easiest to get the main idea in a concrete toy example. Thank you!
> > >
> > > [Generally, I still feel that there are some subtleties and technical details regarding the use of solution functions/reduced form in place of the full SCM that deserve more attention in the main paper. There must be good reasons why SCMs contain internal structure and are not just expressed as a function from the space of exogenous variables to the space of endogenous variables. I think I either fail to see the simplifying assumptions/steps or the authors have had some profound insights here that are of independent interest; in either case, I think this connection warrants further discussion.]

---

> > > > ### Author Response · Authors · 2022-08-09
> > > > **Thank you for the comments - hopefully we can clarify**
> > > >
> > > > We are very thankful for your detailed comments and the opportunity to explain better than we've done until now. Clearly, we are using some non-standard methodology that we should elaborate on in more detail in the paper. In the final version of this work, we will include a more comprehensive explanation. We'd be happy to use the example you provided as part of that exposition.
> > > >
> > > > ### Explicit example
> > > > We are given the solution function
> > > > $$s: \mathbb{R}^3 \to \mathbb{R}^3 : \epsilon \mapsto z = \begin{pmatrix}1 & 0 & 0 \\\\ 1 & 1 & 0 \\\\ 2 & 1 & 1\end{pmatrix}\epsilon$$
> > > >
> > > > Now from this function, we can reconstruct the SCM $z_1=f_1(;\epsilon_1), z_2=f_2(z_1; \epsilon_2), z_3=f_3(z_1, z_2; \epsilon_3)$ in the following way.
> > > >
> > > >
> > > > Looking at the Jacobian (here equal to the matrix $s$), we find that the ordering $(1,2,3)$ makes the Jacobian triangular, so can solve in that order.
> > > >
> > > > We are assuming that any causal mechanism is a bijective function from the noise, conditional on the parents. For simplicity, we write as if the causal mechanisms are functions of all preceeding nodes in the topological ordering and find them to be constant in all non-parents.
> > > > As a general approach:
> > > > - For the first node, we note that $z_1=f_1(\epsilon_1)=s_1(\epsilon_1)$.
> > > > - For the second node, we write $z_2 = f_2(z_1; \epsilon_2) = s_2(f^{-1}_1(z_1); \epsilon_2)$.
> > > > - For the third node, we write $z_3= f_3(z_1,z_2; \epsilon_3) = s_3(f^{-1}_1(z_1), f^{-1}_2(z_1; z_2); \epsilon_3)$.
> > > > - For the $n$th node, we write $z\_n= f\_n(z\_1,...,z\_{n-1}; \epsilon\_n) = s_n(f^{-1}\_1(z\_1), ..., f^{-1}\_{n-1}(z\_1, ..., z\_{n-2}; z\_{n-1}); \epsilon\_n)$.
> > > >
> > > > In the present example, we infer:
> > > > - $z_1 = f_1(\epsilon_1)=\epsilon_1$. Note that the inverse is $\epsilon_1 = f_1^{-1}(z_1)=z_1$
> > > > - $z_2 = f_2(z_1; \epsilon_2) = s_2(f^{-1}_1(z_1); \epsilon_2)=s_2(z_1; \epsilon_2)=z_1 + \epsilon_2$. Note that the inverse is $\epsilon_2 = f_2^{-1}(z_1; z_2)=z_2 - z_1$
> > > > - \begin{align*}z_3&= f_3(z_1,z_2; \epsilon_3) =  s_3(f^{-1}_1(z_1), f^{-1}_2(z_1; z_2); \epsilon_3)=s_3(z_1, z_2 - z_1; \epsilon_3) \\\\ &=2 z_1 + (z_2 - z_1) + \epsilon_3 = z_1 + z_2 + \epsilon_3 \end{align*}
> > > >
> > > > Hence, we've recovered the causal mechanisms $f_1, f_2, f_3$ just from the functional form of $s$.
> > > >
> > > > Subsequently, we can do counterfactual inference. We observe $z=(1,2,4)$, use the inverse solution function $s^{-1}$ to obtain $\epsilon=(1, 1, 1)$. Then, we do an intervention $do(z_2=0)$. We can model this in terms of an intervened noise space $\tilde \epsilon$, for example via an intervened causal mechanism $\tilde z_2 = \tilde f_2(\tilde \epsilon_2)=\tilde \epsilon_2$ and are then interested in the intervention $\tilde \epsilon_2 = 0$. For all the non-intervened variables, the causal mechanism and the noise variable are unchanged, so $\tilde f_1 =f_1,\tilde f_3=f_3, \tilde \epsilon_1=\epsilon_1, \tilde\epsilon_2 = \epsilon_2$, to get $\tilde \epsilon = (1, 0, 1)$.
> > > >
> > > > We unroll the interventional mechanisms $\tilde f_1, \tilde f_2, \tilde f_3$, to get interventional solution function
> > > > $$
> > > > \tilde s_{I=2} : \mathbb{R}^3 \to \mathbb{R}^3 :  \tilde \epsilon \mapsto \tilde z = \begin{pmatrix}1 & 0 & 0 \\\\ 0 & 1 & 0 \\\\ 1 & 1 & 1\end{pmatrix}\tilde \epsilon
> > > > $$
> > > > We evaluate this to get $\tilde z = \tilde s_{I=2}(\tilde \epsilon)=(1, 0, 2)$, as expected.
> > > >
> > > > ### Bigger picture
> > > > In our ILCM model, we are able to identify the solution function from the weakly supervised data. With our assumptions, in particular the conditional invertibility of the causal mechanisms, the SCM can be recovered from the solution function / reduced form SCM. This allows us to conduct interventions, as shown in our paper in Figures 4 and 5.
> > > >
> > > > Does this clarify the methodology?

---

> > > > > ### Author Response · Authors · 2022-08-09
> > > > > **Additional comment**
> > > > >
> > > > > We would like to add two more comments.
> > > > >
> > > > > First, we describe how we extract causal mechanisms and graphs in Section C.4 of the supplementary material. The heuristic algorithm ("ILCM-H") corresponds to the iterative procedure we described in our previous reply. In the final version of our manuscript, we would describe this in more detail and more prominently in the main paper (especially since we will then have an extra page at our disposal).
> > > > >
> > > > > Second, we would like to point out that in practice from pixels we can generate causal variables, infer intervention targets, and generate observational, interventional, and counterfactual samples (in data space) from ILCMs without extracting the causal mechanisms $f_i$ as described above.
> > > > > For instance, given one sample $x$ and an intervention target $I=\{i\}$, we can sample from counterfactual distributions $p(\tilde{x}|x, I)$ by encoding $x$ to $e \sim q(e|x)$, sampling $\tilde z_i \sim \mathcal{N}$ (or setting to a desired value), setting $\tilde e_i = s_i^{-1}(e_{\setminus i}; \tilde z_i)$  , keeping $\tilde e_{\setminus i} = e_{\setminus i}$, and decoding $\tilde{x} \sim p(\tilde x|\tilde{e})$. This is what we do in our visualizations.
> > > > >
> > > > > The cases that require extraction of the causal mechanisms are:
> > > > > - Inferring the graph with the heuristic ILCM-H method
> > > > > - Inferring interventions / counterfactuals from causal variables, as this requires us to invert the solution function $s$, which first requires computing the mechanisms.

---

> > > > > ### Comment · Reviewer_GuNY · 2022-08-09
> > > > > **Thanks for the clarification**
> > > > >
> > > > > I thank the authors for responding and explaining their method in the context of my toy example; this was certainly helpful, and I understand it better now. It would be helpful to include something along those lines as suggested.
> > > > >
> > > > > I am intrigued and would like to better understand how general this procedure is beyond the considered setting: are there types of interventions for which this does not work? how about causally insufficient systems / dependent noises? how restrictive is the assumption of real-valued variables and bijectivity etc...
> > > > >
> > > > > But I understand that this is outside the scope of the current paper, certainly outside the scope of the discussion period, and I do not expect to authors to still reply to this. I will still update my review and tend to increase my score.

---

> > > > > > ### Author Response · Authors · 2022-08-09
> > > > > > **Thanks for the fruitful discussion**
> > > > > >
> > > > > > Thank you for the reply, we are glad to hear that we were able to clarify our approach.
> > > > > >
> > > > > > Your questions and the toy example were very helpful. In addition to improving our description of ILCMs along these lines, in the final version of our paper we will also discuss which of our assumptions are required for reduced-form SCMs to capture equivalent information to SCMs, and for ILCMs to be equivalent to ELCMs.

---

### Author Response · Authors · 2022-08-02
**Overall response**

We thank all the reviewers for their thoughtful comments and constructive feedback. We are encouraged that the reviewers find the problem of causal representation learning relevant (GuNY, UyEP) and appreciate that we prove identifiability in the weakly supervised setting (GuNY, eczw, HAYV). In addition, we provide a novel learning strategy to infer causal structure without explicitly forming graphs, which the reviewers considered one of the strengths of the paper (GuNY, eczw, HAYV). Some reviewers also "enjoyed reading the paper" (HAYV) and found it "well-written and well-motivated" (GuNY).

The reviewers also gave some excellent suggestions for improvement. Reviewer eczw has some concerns regarding the scale of the experiments. We attempted to address this by including an additional experiment, in which we train our model on a synthetic task with a varying number of variables. We found that our method scales well to 10 causal variables without any additional tuning. Furthermore, we have improved the writing in a new version to address some concerns the reviewers had.

Reviewers eczw and HAYV correctly note that we need to make limiting assumptions to arrive at our theoretical result. However, it is well known that some additional assumptions are necessary for causal identifiability. We believe that we increase the understanding of what kind of assumptions are sufficient, and provide a practical method within our regime. Hence, we think that our paper makes an important contribution to causal representation learning.

---

> ### Author Response · Authors · 2022-08-09
> **Any more questions?**
>
> We would like to thank all reviewers again for their helpful comments. We hope that we were able to address all points with our replies and the updated paper.
>
> If there are any open questions, please let us know. We would be more than glad to discuss them before the end of the discussion period later today.

---

### Comment · Area_Chair_RCH2 · 2022-08-07
**Discussion with Authors**

Dear Reviewers! Thank you so much for your time on this paper so far.

The authors have written a detailed response to your concerns. How does this change your review?

Please engage with the authors in the way that you would like reviewers to engage your submitted papers: critically and open to changing your mind.

Looking forward to the discussion!

---

### Meta-Review · Area_Chair_RCH2 · 2022-08-26

**Recommendation:** Accept
**Confidence:** Certain

**Metareview:**

The reviewers were split about this paper: on one hand they would have liked to see better experimental results, particularly for larger graphs, on the other they appreciated the identifiability results and the ILCM algorithm. After going through the paper and discussion I have voted to accept for the following reason: even though the experimental results could be strengthened, papers with novel approaches to long-standing problems are the kind that make NeurIPS an uniquely interesting conference, particularly if those paper have strong theoretical guarantees. I urge the authors to take all of the reviewers changes into account (if not already done so). Once done this paper will be a nice addition to the conference!

**Award:**

No

---

### Decision · Program_Chairs · 2022-09-14

Accept